# AIPO: Agreement-Aware Iterative Preference Optimization for Length Exploitation Mitigation

## Abstract

Direct Preference Optimization (DPO) is gaining popularity as an alternative to Proximal Policy Optimization (PPO) for aligning Large Language Models (LLMs). Recent research on aligning LLMs iteratively with synthetic or partially synthetic data has shown promising outcomes, facilitating the scalability of DPO training in both academic settings and proprietary models such as Llama 3. Despite its success, we observe that the issue of length exploitation in DPO becomes more pronounced during iterative preference optimization, with the severity escalating progressively with each iteration. This observation prompts an in-depth examination of iterative preference optimization with synthetic data. In this paper, we present our findings and analyses in building our iterative preference optimization pipeline. Specifically, we analyze the issue of length exploitation in this iterative process and propose a novel training objective for iterative preference optimization, namely **A**greement-aware **I**terative **P**reference **O**ptimization (AIPO). To demonstrate the effectiveness of our proposed method, we conduct extensive experiments and show that it achieves state-of-the-art performance on MT-Bench, AlpacaEval 2.0, and Arena-Hard.

## 1 Introduction

Reinforcement Learning from Human Feedback (RLHF) (Christiano et al., 2017; Stiennon et al., 2020a) has emerged as a crucial technique to align Large Language Models (LLMs) with human preferences. Although RLHF is effective compared to Supervised Fine-Tuning (SFT) (Ouyang et al., 2022a), it encounters scalability challenges due to its training inefficiency and multistage process. Recently, Direct Preference Optimization (DPO) has gained attention due to its scalability to large-scale models and its superior performance compared to Proximal Policy Optimization (PPO), thus serving as a good alternative to the conventional RLHF pipeline (Dubey et al., 2024). The key success of DPO derives from re-parameterizing the reward model using optimal policy obtained from the reinforcement learning phase, enabling direct training of the language model via reward modeling. Consequently, DPO facilitates efficient scaling for training large-scale models to learn human feedback. Nonetheless, DPO still faces challenges due to the labor intensive labeling process required for preference data collection. Currently, advances in both proprietary and open-source LLMs have demonstrated human-level performance across various tasks (Dubey et al., 2024; Achiam et al., 2023; Yang et al., 2024), indicating their potential to autonomously generate preference data. Based on this fact, replacing human annotation with LLM-generated data becomes a popular solution to the aforementioned scalability challenge.

Recent studies (Yuan et al., 2024; Wu et al., 2024a; Tran et al., 2023; Chen et al., 2024) show that aligning LLMs with synthetic data in an iterative manner can effectively achieve continuous improvement in performance and allow for a higher performance ceiling. However, despite their success, the length exploitation issue that exists in the generic DPO setting (Park et al., 2024) has a strong impact on performance, as observed in our research and noted in recent works (Yuan et al., 2024; Tran et al., 2023). Current benchmarks for preference optimization exhibit a common bias toward lengthy responses, which are less efficient for users to consume and require more hardware resources to generate. Additionally, creating and training on synthetic data for lengthy responses consume more hardware resources. Therefore, we argue that high scores on existing benchmarks are

insufficient to reflect alignment performance accurately, and the length exploitation issue needs more attention. Moreover, due to the complexity of multistage iterative training, numerous combinations of training procedures remain unexplored. These include the detailed steps for creating synthetic preference pairs, the data amount to be trained on each iteration, the tuning of hyperparameters for each training stage, data selection and cleaning, and the combination with other training methods, etc. Despite the success of existing works, these questions remain unanswered. We thus argue that research in the area of iterative preference optimization is still in its early stages and that the fundamental building blocks for iterative preference optimization are still under explored.

In this work, we showcase our training recipe for aligning LLMs with purely synthetic data iteratively by carefully examining the design choices for each component in iterative preference optimization, serving as a good starting point for investigating iterative preference optimization. During this process, we reveal the severe length issue in iterative preference optimization, which we believe significantly limits the potential application of this method. Through our analysis, we propose solutions to overcome the length issue in training stages and introduce our own training objective for iterative preference optimization. Our contributions can be summarized as follows:

- **Data: a Synthetic Data Curation Pipeline for Preference Optimization**. We examine the validity of preference optimization with synthetically generated data. The pipeline includes instruction creation, response generation, and preference ranking. We conclude that models trained with synthetic data yield better performance and have the potential to scale up at a low cost. §3.1
- **Finding: Length Exploitation Issue in Iterative Training Strategy**. We define our iterative preference optimization training strategy and perform ablations in different configurations. We observe a more severe length exploitation issue during iterative training with synthetic data. §3.2
- **AIPO: Optimized Training Objective for Iterative Preference Optimization**. We dive deep into the length exploitation issue and discover that one of the potential causes is related to the DPO loss. To remedy this, we introduce a new optimized training objective, AIPO, which is more suitable for iterative preference optimization scenarios. §4

Altogether, we propose an effective training recipe for iterative preference optimization, including the AIPO training objective for iterative training. By leveraging this new training recipe, we achieve state-of-the-art performance on benchmarks including MT-Bench, AlpacaEval 2.0 and Area-Hard.

## 2 PRELIMINARIES AND RELATED WORK

### 2.1 DIRECT PREFERENCE OPTIMIZATION

Direct Preference Optimization (DPO) (Rafailov et al., 2024) is derived from the reinforcement learning (RL) phase of the RLHF pipeline (Thoppilan et al., 2022; Stiennon et al., 2020b; Bai et al., 2022; Ouyang et al., 2022b). The objective of the RL phase is formulated as follows:

$$\max_{\pi_\theta} \mathbb{E}_{x \sim \mathcal{D}, y \sim \pi_\theta(y|x)} \left[ r_\phi(x, y) \right] - \beta \mathbb{D}_{\text{KL}} \left[ \pi_\theta(y \mid x) \| \pi_{\text{ref}}(y \mid x) \right], \quad (1)$$

where $\pi_\theta$ is the policy model, $\pi_{\text{ref}}$ is the reference policy, $r_\phi$ is the reward model, and $\beta$ is a hyperparameter to control the deviation from the reference policy. Instead of training an explicit reward model and employing RL, DPO reparameterizes the reward utilizing an implicit optimal reward function:

$$r(x, y) = \beta \log \frac{\pi_\theta(y|x)}{\pi_{\text{ref}}(y|x)} + \beta \log Z(x), \quad (2)$$

where $Z(x)$ is the partition function. Incorporating the reward function into the Bradley-Terry model (Bradley & Terry, 1952),

$$p(y_w \succ y_l | x) = \sigma(r(x, y_w) - r(x, y_l)), \quad (3)$$

cancels the partition function and yield the DPO training objective:

$$\mathcal{L}_{\text{DPO}} \left( \pi_\theta; \pi_{\text{ref}} \right) = -\mathbb{E}_{(x, y_w, y_l) \sim \mathcal{D}} \left[ \log \sigma \left( \beta \log \frac{\pi_\theta \left( y_w \mid x \right)}{\pi_{\text{ref}} \left( y_w \mid x \right)} - \beta \log \frac{\pi_\theta \left( y_l \mid x \right)}{\pi_{\text{ref}} \left( y_l \mid x \right)} \right) \right], \quad (4)$$

where $y_w$ and $y_l$ are the chosen and rejected responses, respectively. Consequently, DPO training can be directly applied to binarized preference datasets, which include ternary preference pairs $(x, y_w, y_l)$. DPO eliminates the need for an explicit reward model and RL during training, making it more suitable for scaling up the RLHF training stage.

## 2.2 PREFERENCE OPTIMIZATION OBJECTIVES

Several preference optimization objectives have been developed in addition to DPO. One line of research is to study preference optimization without relying on a reference model (Xu et al., 2023; Hong et al., 2024; Meng et al., 2024). Other methods try to add a margin between the chosen and rejected responses (Gheshlaghi Azar et al., 2023; Zhao et al., 2023; Zheng et al., 2023a). R-DPO (Park et al., 2024) and SimPO (Meng et al., 2024) also explore length-controlled approaches by leveraging length regularization and length normalization, respectively. RPO (Liu et al., 2024) proposes a training objective that incorporates a weighted SFT loss as the regularization term. It is worth noting that, despite the effectiveness of certain methods, the validation of the majority of these improved DPO losses has been primarily limited to non-iterative settings, with a lack of validation in iterative contexts.

## 2.3 ITERATIVE ALIGNMENT METHODS

Iterative alignment is gaining popularity for achieving continuous improvement through successive training iterations. Self-Rewarding (Yuan et al., 2024) focuses on self-involvement of large language models, utilizing the LLM-as-a-Judge mechanism (Zheng et al., 2023b) to score their own responses, thus mitigating the performance bottlenecks that can arise from a frozen judge model. It leverages iterative training to generate self-judgement using an up-to-date model. Based on Self-Rewarding, Iterative RPO (Pang et al., 2024) aims to improve reasoning ability through iterative preference optimization by utilizing Chain-of-Thought (CoT) (Wu et al., 2023) reasoning. Meta-Rewarding (Wu et al., 2024a) focuses on improving the self-judging ability in self-rewarding by adding a role of meta-judge to judge the model's own judgement. sDPO (Kim et al., 2024) suggests dividing the available preference datasets into multiple subsets and training on each subset iteratively. Snorkel-Mistral-PairRM-DPO (Tran et al., 2023) is trained iteratively, starting from an initial prompt pool sampled from UltraFeedback. The model is prompted with instructions from UltraFeedback to generate several candidate responses and then uses PairRM (Jiang et al., 2023b) as reward model to rank the responses. Finally, it trains on the top and bottom responses with DPO. SPPO (Wu et al., 2024b) approximates the Nash equilibrium iteratively by pushing the chosen rewards to be close to 1/2 and the rejected rewards to be -1/2. Although all these methods involve iterative training, they do not explain the differences between non-iterative and iterative preference optimization. In addition, they lack a detailed analysis of the design choices and properties involved in the iterative preference optimization.

# 3 ITERATIVE PREFERENCE OPTIMIZATION WITH SYNTHETIC DATA

In this section, we detail the step-by-step development of a state-of-the-art training recipe for iterative preference optimization, examining the design choices for each component. We start with the non-iterative baseline, which is trained on the existing pairwise preference dataset, and then move on to synthetic preference pairs and iterative training. Finally, we present our refined training recipe for subsequent experiments and emphasizing the challenges posed by the iterative training approach.

## 3.1 SYNTHETIC DATA CURATION

The training data for DPO consists of a large number of preference pairs $(x, y_w, y_l)$. Previous works (Yuan et al., 2024; Wu et al., 2024a; Tran et al., 2023; Chen et al., 2024) have suggested various methods for creating synthetic preference pairs for preference optimization, but there is a lack of detailed comparison. To study the roles and effects of different components in the data curation pipeline within preference optimization, we conducted a thorough analysis of all aspects, including instructions, responses, and preference rankings.

**Self-Generated Responses vs. External Model Responses**. Recent works (Tran et al., 2023; Meng et al., 2024; Wu et al., 2024b; Yuan et al., 2024; Pang et al., 2024; Wu et al., 2024a) suggest leveraging self-generated responses and existing state-of-the-art reward models to build preference pairs, despite the presence of existing responses and rewards in preference datasets. To explore this difference, we conduct experiments on the UltraFeedback dataset. We begin with vanilla DPO training by training for a single epoch on the 60K preference pairs from UltraFeedback Binarized. We

Table 1: The ablation of the synthetic preference pairs for DPO training. UF indicates the instructions and responses from UltraFeedback, SI(·) indicates generating self instruct based on the inputs, Gen(·) indicates generating candidate responses with policy model by taking inputs as prompts, and PairRM(·) indicates ranking responses by PairRM.

| | Training Data | | Arena-Hard | | AlpacaEval 2.0 | | | MT-Bench |
|---|---|---|---|---|---|---|---|---|
| | Instruction | Response | WR (%) | Avg. Token | LC (%) | WR (%) | Avg. Len | GPT-4-Turbo |
| (1) | UF | UF | 14.4 | 535 | 20.4 | 16.5 | 1664 | 6.3 |
| (2) | UF | PairRM(UF) | 13.6 | 512 | 20.8 | 16.4 | 1623 | 6.2 |
| (3) | UF | PairRM(Gen(UF)) | 17.6 | 649 | 23.7 | 25.2 | 2198 | **6.5** |
| (4) | SI(UF) | PairRM(Gen(SI(UF))) | **19.6** | 615 | **26.0** | **28.2** | 2130 | 6.4 |

Table 2: Comparison of methods for ranking candidate responses. We generate candidate responses for 60K instructions from UltraFeedback. For LLM judge, we use the LLM-as-a-Judge prompt from Self-Rewarding and employ Mixtral-8x7B-Instruct-v0.1 as the external LLM judge model.

| Response Ranking | Arena-Hard | | AlpacaEval 2.0 | | | MT-Bench |
|---|---|---|---|---|---|---|
| | WR (%) | Avg. Token | LC (%) | WR (%) | Avg. Len | GPT-4-Turbo |
| Self-Reward | 14.5 | 729 | 21.0 | 22.5 | 2225 | 6.1 |
| External LLM | 13.9 | 603 | 20.5 | 19.6 | 1890 | 6.1 |
| PairRM | **17.6** | 649 | **23.7** | **25.2** | 2198 | **6.5** |

then replace GPT-4 with PairRM as the ranking method for candidate responses to ensure a fair comparison. Finally, we replace existing responses with self-generated ones, using PairRM to rank them and select the best and worst responses as $y_w$ and $y_l$, following Snorkel-Mistral-PairRM-DPO (Tran et al., 2023). The results in Tab. 1 show that replacing GPT-4 annotations with PairRM rankings (row (1) vs. row (2)) is not crucial for performance, while the use of self-generated responses (row (3)) contributes the most to the performance gap. We also highlight that the average response length significantly increases when using self-generated responses (row (3) and (4)) compared to externally-generated ones.

**Synthetic Instructions vs. Human Instructions** As suggested in Self-Rewarding (Yuan et al., 2024), we employ Self-Instruct (Wang et al., 2023) to build a fully synthetic training pipeline, starting from synthetic instructions. Subsequently, the same pipeline is used to generate candidate responses and rank them using PairRM, as mentioned above. As shown in Tab. 1 (row 4), training with fully synthetic instructions generated by self-instruct achieves performance competitive to human instructions in UltraFeedback.

**Reward Models vs. LLM Judges** Existing works primarily use two methods to rank candidate responses: 1) using pre-existing reward models (e.g., PairRM) (Meng et al., 2024; Park et al., 2024; Tran et al., 2023), and 2) prompting LLMs with a judge prompt to score responses (Yuan et al., 2024; Pang et al., 2024; Wu et al., 2024a). The LLM judge model can either be external or the LLM itself. Although this novel approach shows promise in incorporating self-involvement in iterative training, our experimental results reveal that LLM judges underperform compared to dedicated reward models such as PairRM. As shown in Tab. 2, despite having only 0.4B parameters, PairRM outperforms the LLM judge across all benchmarks. Moreover, prompting LLMs to generate scores and rank responses is computationally expensive. Thus, we opted for pre-existing reward models like PairRM to ensure simplicity and optimal performance. We also note that the reward model largely determines the upper limit of preference optimization, and a stronger reward model can further improve performance (Wu et al., 2024a). We leave this analysis for future work.

## 3.2 TOWARDS ITERATIVE PREFERENCE OPTIMIZATION

Through the detailed comparative experiments described above, we have developed our synthetic data curation pipeline by integrating the optimal design choices identified through ablation studies, comprising: (1) creating synthetic instructions using self-instruction, (2) generating candidate responses using the model from the current iteration, and (3) ranking the responses with an existing state-of-the-art reward model. We emphasize that this pipeline is entirely synthetic: starting from an initial prompt pool, we are capable of generating a substantial amount of data until the model

---

**Algorithm 1:** Iterative Training Pipeline.

---

**Input:** $X^{\text{pool}}$: Initial prompt pool, $\theta_0$: Base model, $T$: Number of iterations, $P$: Number of new
 instructions, $N$: Number of candidate responses.
**for** $t = 0, \ldots, T - 1$ **do**
    **for** $i = 1, \ldots, P$ **do**
        Generate new instruction: $x^i = \text{SelfInstruct}_{\theta_t}(X^{\text{pool}})$.
        **for** $j = 1, \ldots, N$ **do**
            Generate candidate responses: $y_j^i \sim p_{\theta_t}(\cdot \mid x^i)$.
        **end**
        Rank responses and obtain preference pairs: $(x^i, y_w^i, y_l^i) = \text{PairRM}(x^i, y_1^i, \ldots, y_N^i)$.
    **end**
    Update model weights: $\theta_{t+1} = \arg\min_{\theta} \sum_{i=1}^{P} \mathcal{L}_{\text{AIPO}}(x^i, y_w^i, y_l^i, \theta_t, \theta)$
**end**
**Output:** $\theta_T$

---

converges. Utilizing our synthetic data curation pipeline, we have extended the alignment process in an iterative manner, as delineated in Algorithm 1.

To investigate the properties and impact of iterative training on performance, we maintain a constant total volume of training data while varying the data size per iteration. The results, as shown in Fig. 1, reveal that increasing the number of iterations with smaller data sizes per iteration leads to a higher performance ceiling, highlighting the importance of iterative training. We hypothesize that iterative training improves preference optimization by updating both the data generation model and the reference model for the training objective in each iteration, providing timely feedback. However, using smaller data sizes per iteration requires frequent transitions between training phases, which demands a more sophisticated implementation to maintain training efficiency. Moreover, we observe that further reducing the number of training data per iteration yields only marginal improvements. Therefore, in subsequent experiments, we set the iterative training parameters to 20K preference pairs per iteration, with a default batch size of 256.

Compared to the non-iterative preference optimization presented in Tab. 1, we emphasize that, despite performance improvements, iterative preference optimization leads to server length exploitation. Our objective is to continuously improve performance through iterative preference optimization. Although this approach effectively achieves a high win rate across benchmarks, the issue of length exploitation is particularly evident in this scenario: increased length results in significantly longer training times, ultimately causing performance degradation due to excessive verbosity. To address this apparent and critical issue, we begin by analyzing the problem of length exploitation in iterative training and have managed to resolve it from the perspective of the training objective, as detailed in the following section.

## 4 REVISITING TRAINING OBJECTIVES

### 4.1 ANALYZING DPO IN ITERATIVE TRAINING

Our experiments reveal a significant length exploitation issue in iterative DPO when using self-generated responses, prompting us to investigate the root cause of this undesired behavior. One advantage of using self-generated responses is that they are generated directly by the model produced in the most recent iteration. They represent the model's best capability in following the provided instructions and both the quality and format of the responses generated for each instruction are very similar. To verify this, we analyze the similarity and log probabilities of the responses from different sources, as shown in Tab. 3. Compared to the externally generated responses, the self-generated ones have 1) a much higher average value of the log probabilities for both chosen and rejected responses and 2) a significantly higher similarity between the chosen responses and the rejected responses. When combined with iterative training, in each iteration, new candidate responses are generated using the latest model checkpoint from the previous iteration, and the reference model is also updated. This process makes the training resemble online RLHF to some extent, wherein the LLM receives semi-real-time AI feedback from the reward model based on its generated completions. We hypoth-

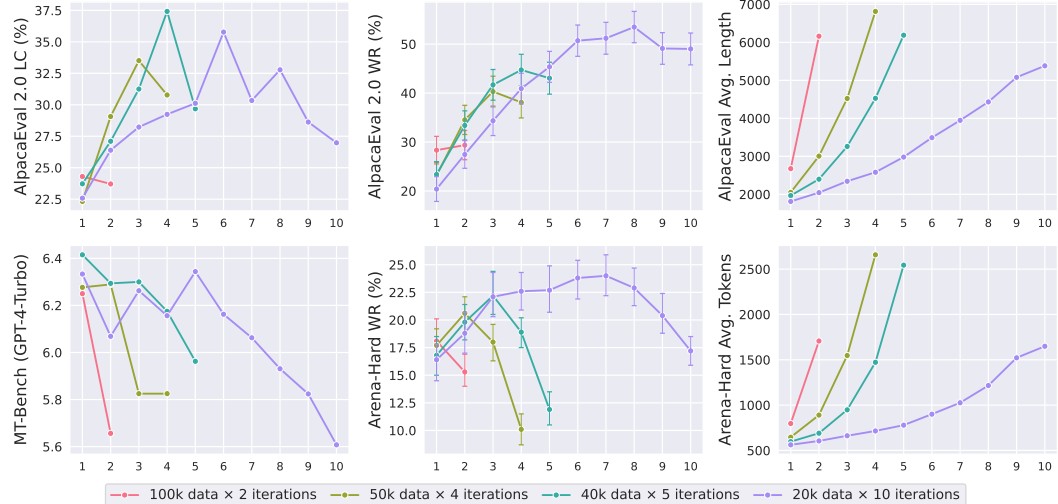

Figure 1: Ablation of iterative training. The horizontal axis represents the training iterations. We train for $T$ iterations, generating $P$ preference pairs in each iteration. In this ablation, we ensure that the number of total generated pairs $T \times P$ remains constant.

esize that this is the primary reason iterative training performs better than non-iterative training, as demonstrated in Sec. 3.2.

However, the similarity of self-generated responses, as indicated in Tab. 3, poses a challenge in distinguishing their quality. Since the candidate responses are similar to each other, it becomes difficult for the reward model to rank them accurately. Moreover, the training objective in Eq. 4 aims to increase the log-likelihood of the chosen responses and decrease that of the rejected responses, while ignoring the intrinsic relationship between them. Forcing the model to distinguish between very similar chosen and rejected responses with high log probabilities can lead to an overestimated gradient value. We hypothesize this makes the DPO training objective susceptible to self-generated preference pairs, consequently degrading the model's learning and resulting in responses that are lengthy and less informative.

## 4.2 AIPO: Agreement-Aware Iterative Preference Optimization

We propose to address the difficulty of learning from self-generated responses by leveraging the feedback from the reference model. To achieve this, we first rewrite the DPO training objective in Eq. 4 as:

$$\mathcal{L}_{\text{DPO}}(\pi_\theta; \pi_{\text{ref}}) = -\mathbb{E}_{(x,y_w,y_l)\sim\mathcal{D}}\left[\log \sigma\left(\beta\left(s_\theta - s_{\text{ref}}\right)\right)\right], \tag{5}$$

where $s_\theta = \log \frac{\pi_\theta(y_w|x)}{\pi_\theta(y_l|x)}$ and $s_{\text{ref}} = \log \frac{\pi_{\text{ref}}(y_w|x)}{\pi_{\text{ref}}(y_l|x)}$. We note that $s_{\text{ref}}$ represents the extent to which the reference model tends to generate the chosen response $y_w$ with a higher probability than the rejected response $y_l$, i.e., the agreement between the reference model and the reward model. We introduce an additional coefficient $\alpha$ to $s_{\text{ref}}$ in Eq. 5. The new training objective is defined as:

$$\mathcal{L}_{\alpha\text{-DPO}}(\pi_\theta; \pi_{\text{ref}}) = -\mathbb{E}_{(x,y_w,y_l)\sim\mathcal{D}}\left[\log \sigma\left(\beta\left(s_\theta - (1+\alpha)s_{\text{ref}}\right)\right)\right], \tag{6}$$

where we set $\alpha > 0$. We note that $s_{\text{ref}}$ is not related to the policy model $\pi_\theta$, thus it is equivalent to adding an additional dynamic target reward margin term to the Bradley-Terry model in Eq. 3, which can be written as:

$$p(y_w \succ y_l|x) = \sigma\left(r(x, y_w) - r(x, y_l) - \alpha\beta \cdot s_{\text{ref}}\right), \tag{7}$$

where $\alpha\beta \cdot s_{\text{ref}}$ is a dynamic target margin for adjusting the distribution of reward margin. As investigated in previous works (Meng et al., 2024), a larger target margin value produces a larger reward margin by flattening the reward difference distribution. Intuitively, $\alpha\beta \cdot s_{\text{ref}}$ resulting in a larger reward margin when the preference of the reference model agrees with that of the reward model according to the selected chosen and rejected responses, and pose resistance when there is a preference mismatch between the reference model and the reward model. This ensures a scaling of

Table 3: The similarity and log probabilities of chosen and rejected responses. We use Sentence Transformers[1] to calculate similarity, and Mistral-7B-Instruct-v0.2 to generate responses and compute the log probabilities. We also include the length-normalized log probabilities in parentheses.

| Type of Responses | Sentence Similarity | Log Probabilities | |
| --- | --- | --- | --- |
| | | Chosen | Rejected |
| Externally-Generated (UltraFeedback) | 0.64 | -361.5 (-1.140) | -461.1 (-1.927) |
| Self-Generated | 0.86 | -107.8 (-0.268) | -110.4 (-0.271) |

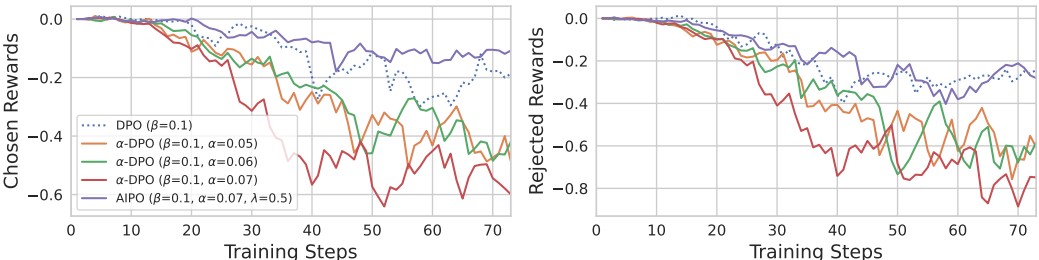

Figure 2: Chosen and rejected rewards during training. The chosen and rejected rewards are drawn from the first iteration of each method.

rewards by considering the agreement between the reference model and the reward model, which helps eliminate the aforementioned problem of using self-generated responses. Next, we analyze the gradient of $\mathcal{L}_{\alpha\text{-DPO}}$. The gradient with respect to $\theta$ can be written as:

$$\nabla_\theta \mathcal{L}_{\alpha\text{-DPO}}(\pi_\theta; \pi_{\text{ref}}) = -\beta \mathbb{E}_{(x,y_w,y_l)\sim\mathcal{D}}\left[ w_\theta \cdot \left( \underbrace{\nabla_\theta \log \pi(y_w \mid x)}_{\text{increase likelihood of } y_w} - \underbrace{\nabla_\theta \log \pi(y_l \mid x)}_{\text{decrease likelihood of } y_l} \right) \right], \quad (8)$$

where

$$w_\theta = \sigma\Big(\beta\big(\underbrace{s_{\text{ref}} - s_\theta}_{\substack{\text{weighted by}\\\text{reward estimate}}} + \underbrace{\alpha \cdot s_{\text{ref}}}_{\substack{\text{weighted by}\\\text{agreement}}}\big)\Big) \quad (9)$$

is the gradient weight. The gradient of the loss function $\mathcal{L}_{\alpha\text{-DPO}}$ preserves the core properties of DPO: it increases the likelihood of preferred responses $y_w$ and decreases the likelihood of dispreferred responses $y_l$, weighted by the reward estimate $s_{\text{ref}} - s_\theta$. Importantly, $\alpha$-DPO adds an additional weighting term $\alpha\beta \cdot s_{\text{ref}}$, which weights the gradient by how much more the reference model prefers the chosen response over the rejected response, i.e., the agreement between the preferences of the reference model and the reward model.

In Fig. 2, we investigate the trend of chosen and rejected rewards during training with $\alpha$-DPO. The results show that $\alpha$-DPO leads to a decrease in both chosen and rejected response log probabilities, which might be harmful as noted in previous work (Pang et al., 2024). However, it is interesting that the performance of $\alpha$-DPO improves despite the decrease in log probabilities for both the chosen and rejected responses by the policy model. We highlight that the decrease in probabilities of self-generated responses also indicates the shift of policy model's output distribution, reflecting that $\alpha$-DPO provides a clear target for learning preferences, thereby making it easier for the model to learn preferences. Since rapid changes in the output distribution may be unstable, we employ Negative Log Likelihood (NLL) loss as a compensatory measure, following previous works (Pang et al., 2024; Dubey et al., 2024). The NLL term is defined as:

$$\mathcal{L}_{\text{NLL}} = -\frac{1}{|y_w|} \log\big(\pi_\theta(y_w \mid x)\big). \quad (10)$$

Combining $\alpha$-DPO with NLL term, our AIPO training objective for iterative preference optimization is defined as:

$$\mathcal{L}_{\text{AIPO}} = \mathcal{L}_{\alpha\text{-DPO}} + \lambda \cdot \mathcal{L}_{\text{NLL}} \quad (11)$$

where $\lambda$ balances the relative importance of $\mathcal{L}_{\text{NLL}}$.

---

[1]https://sbert.net/

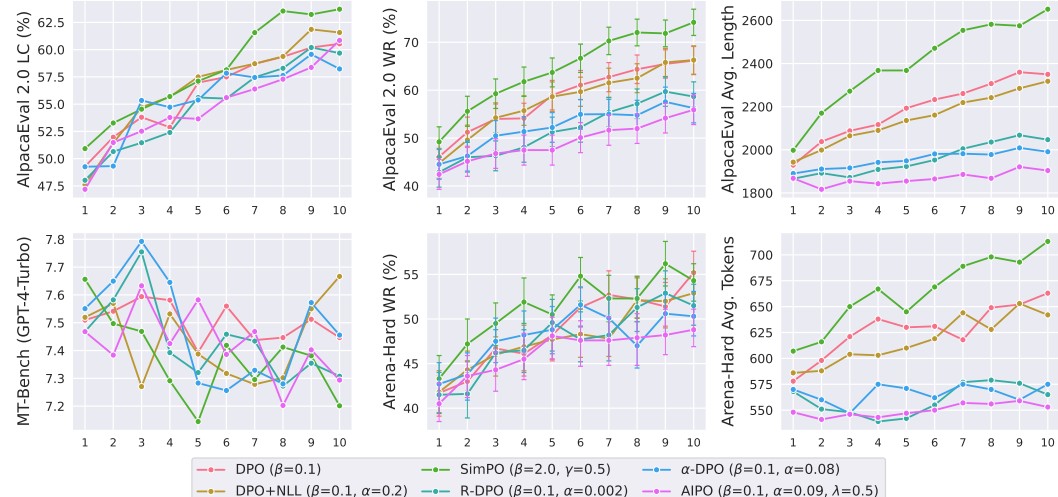

Figure 3: Comparison of different training objectives under iterative alignment setting. The horizontal axis represents the training iterations.

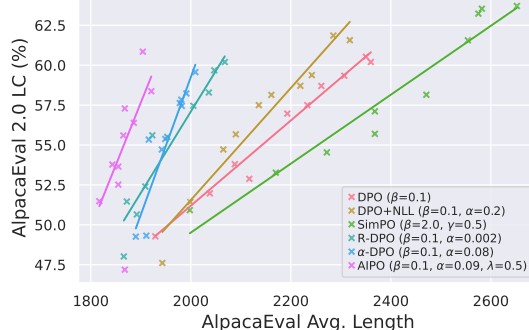

Figure 4: The performance-to-length chart comparing different training objectives, evaluated using length-controlled win rate of AlpacaEval 2.0.

## 5 EXPERIMENTS

### 5.1 EXPERIMENTAL SETUP

Following previous works (Tran et al., 2023; Meng et al., 2024; Wu et al., 2024b), we use Ultra-Feedback (Cui et al., 2023) as the data source for all experiments. We evaluate our model on three benchmarks: MT-Bench (Zheng et al., 2023b), AlpacaEval 2.0 (Li et al., 2023; Dubois et al., 2024), and Arena-Hard (Li et al., 2024), all using GPT-4-Turbo (11/06) as an automatic annotator. Our experiments focus on AlpacaEval 2.0 and Arena-Hard due to their better separability, while MT-Bench results are included for comparison. For more detailed settings, please refer to Sec. A.1 in appendix.

### 5.2 COMPARISONS WITH PREFERENCE OPTIMIZATION METHODS

We conducted extensive experiments to compare various methods within our iterative settings, validating (1) the hyperparameters in vanilla DPO and DPO+NLL, (2) the effectiveness of $\alpha$-DPO in length control, (3) the role of NLL term in training, and (4) the overall performance of AIPO and its generalization to base LLM models. Specifically, we compare our approach to DPO and DPO+NLL, which serve as baselines, SimPO, a state-of-the-art training objective in non-iterative settings, and R-DPO, which is tailored for length control in non-iterative settings. To ensure the stability of the results, we conducted training for a total of 10 iterations, evaluating performance on benchmarks with the consideration of length increment.

**Analysis of DPO and DPO+NLL**. For each method, we perform comprehensive hyperparameter searches to (1) identify the optimal combination of hyperparameters and (2) analyze the attributes

Table 4: The detailed comparison with other iterative preference optimization methods and proprietary models. To ensure a fair comparison with other methods, for iterative training, we select models based on both win rate and response length on AlpacaEval 2.0 and Arena-Hard.

| Methods | Arena-Hard | | AlpacaEval 2.0 | | | MT-Bench |
| | WR (%) | Avg. Token | 2.0 LC (%) | 2.0 WR (%) | Avg. Len | GPT-4-Turbo |
| --- | --- | --- | --- | --- | --- | --- |
| *Baselines* | | | | | | |
| **Mistral-7B-Instruct-v0.2** | 12.8 | 526 | 22.0 | 16.8 | 1610 | 6.2 |
| **Llama-3-8B-Instruct** | 20.7 | 584 | 30.4 | 30.3 | 1955 | 6.8 |
| **Gemma-2-9B-It** | 42.3 | 567 | 48.4 | 34.4 | 1568 | 7.5 |
| *Mistral-7B-Instruct-v0.2, UltraFeedback, PairRM* | | | | | | |
| **Snorkel-Mistral-PairRM-DPO** (Tran et al., 2023) | 20.7 | 564 | 26.4 | 30.2 | 2736 | 6.2 |
| **SPPO** (Wu et al., 2024b) | **21.8** | 572 | **28.5** | **31.0** | 2163 | **6.5** |
| **AIPO** | 16.7 | 468 | 26.2 | 21.3 | 1669 | 6.3 |
| *Llama-3-8B-Instruct, Open Assistant dataset* | | | | | | |
| **Self-Rewarding** (Yuan et al., 2024) | 28.2 | - | 34.9 | 34.6 | 1967 | - |
| **Meta-Rewarding** (Wu et al., 2024a) | 29.1 | - | 39.4 | 39.5 | 2003 | - |
| *Llama-3-8B-Instruct, UltraFeedback, PairRM* | | | | | | |
| **SPPO** (Wu et al., 2024b) | **34.0** | 597 | 38.8 | 39.9 | 2066 | 6.8 |
| **AIPO** | 33.8 | 585 | **46.2** | **45.3** | 1977 | **6.9** |
| *Gemma-2-9B-It, UltraFeedback, PairRM* | | | | | | |
| **SPPO** (Wu et al., 2024b) | 51.1 | 609 | 53.3 | 47.7 | 1803 | 7.6 |
| **AIPO** | **53.0** | 637 | **59.1** | **49.6** | 1809 | 7.6 |

of various methods regarding length-related issues under our iterative settings. Due to space constraints, we present the detailed results in Sec. A.5 of the appendix. Through a thorough ablation study, we conclude that adjusting the value of $\beta$ in DPO and the value of $\alpha$ to control the coefficient of the NLL term in DPO+NLL, while limiting the increase in response length with training iterations, also constrains performance improvements, leaving the performance-to-length ratio unchanged. Thus, we emphasize that merely controlling length growth is insufficient. It is crucial to consider the performance-to-length ratio along with a steeper slope, as depicted in Fig. 4.

$\alpha$-**DPO for length control**. As illustrated in Fig. 3, although the performance of SimPO improves significantly with training iterations, it generates considerably longer responses compared to other methods. This result contradicts the aim of length normalization employed in the SimPO training objective, suggesting that length normalization is not suitable for controlling length exploitation in our iterative settings. Conversely, $\alpha$-DPO demonstrates competitive performance with substantially shorter responses in iterative contexts. Designed specifically for length control, R-DPO achieves performance comparable to $\alpha$-DPO in length control within iterative settings. Fig. 4 provides a clearer visualization of the performance-to-length ratio and demonstrates that $\alpha$-DPO remains superior (with a steeper slope) in mitigating the increase in response length during iterative training.

**NLL term for stable training**. The results further indicate that including the NLL term in the DPO offers only marginal performance improvement, implying that its primary function is to stabilize the distribution of the policy model throughout the training process. However, incorporating the NLL term into $\alpha$-DPO, resulting in AIPO, effectively mitigates the trend of decline in rewards, as demonstrated in Fig. 2, and further improves performance at equivalent length. As shown in Fig. 3, AIPO maintains a nearly constant response length throughout iterative training, albeit with an acceptable reduction in performance compared to DPO and SimPO. From the evaluation on Arena-Hard, it is evident that, compared to R-DPO, AIPO exhibits less variation in response length, indicating greater stability in length control than direct regularization by response length. This underscores the importance of incorporating the NLL term in AIPO to stabilize the output log-likelihood distributions of the winning and losing pairs of the policy model, ultimately leading to better performance.

**AIPO for various LLM base models**. In Tab. 4, we further compare AIPO with other iterative preference optimization methods. Using Mistral-7B-Instruct-v0.2 as the base model, we observe that the performance ceiling is significantly constrained without increasing the response length, underscoring the difficulty of improving the win rate without increasing the response length when the base model's capacity is limited. This suggests a potential issue of length gameability in AlpacaE-

Table 5: Results of our model size scaling experiment and comparison with proprietary models.

| Methods | Parameters | Arena-Hard | | AlpacaEval 2.0 | | | MT-Bench |
|---|---|---|---|---|---|---|---|
| | | WR (%) | Avg. Token | 2.0 LC (%) | 2.0 WR (%) | Avg. Len | GPT-4-Turbo |
| **Mistral-Large-Instruct-2407-AIPO** | 123B | **82.6** (+12.2) | 659 | **63.0** (+6.9) | **66.2** (+22.6) | 2266 | 8.5 (+0.1) |
| **GPT-4-Turbo (04/09)** | - | 82.6 | 662 | 55.0 | 46.1 | 1802 | - |
| **Claude 3.5 Sonnet (06/20)** | - | 79.3 | 567 | 52.4 | 40.6 | 1488 | - |
| **GPT-4 Omni (05/13)** | - | 79.2 | 696 | 57.5 | 51.3 | 1873 | - |
| **GPT-4o Mini** | - | 74.9 | 668 | 50.7 | 44.7 | 1861 | - |
| Mistral-Large-Instruct-2407 | 123B | 70.4 | 623 | 56.1 | 43.6 | 1700 | 8.4 |
| **Llama-3-70B-Instruct-AIPO** | 70B | 63.5 (+16.9) | 616 | 60.5 (+26.1) | 60.1 (+26.9) | 2081 | 8.2 (+0.3) |
| **Gemma-2-27B-It-AIPO** | 27B | 63.5 (+8.3) | 643 | 57.8 (+7.7) | 48.5 (+11.0) | 1768 | 8.0 (+0.3) |
| **Claude 3 Opus (02/29)** | - | 60.4 | 541 | 40.5 | 29.1 | 1388 | - |
| **Gemma-2-27B-It** | 27B | 55.2 | 594 | 50.1 | 37.5 | 1628 | 7.7 |
| **Mistral-Nemo-Instruct-2407-AIPO** | 12B | 51.4 (+12.0) | 551 | 60.3 (+13.9) | 57.5 (+15.8) | 1978 | 7.4 (+0.0) |
| **Claude 3 Sonnet (02/29)** | - | 46.8 | 552 | 34.9 | 25.6 | 1420 | - |
| **Llama-3-70B-Instruct** | 70B | 46.6 | 591 | 34.4 | 33.2 | 1919 | 7.9 |
| **Mistral-Nemo-Instruct-2407** | 12B | 39.4 | 556 | 46.4 | 41.7 | 1883 | 7.4 |
| **GPT-4 (06/13)** | - | 37.9 | 354 | 30.2 | 15.8 | 1140 | - |
| **GPT-3.5 Turbo (06/13)** | - | 24.8 | 401 | 22.7 | 14.1 | 1328 | - |

val 2.0 and Arena-Hard. Despite this, we achieve competitive performance without increasing the response length, underlining the short response length of our method. Conversely, training on robust LLMs (e.g., Llama-3-8B-Instruct) tends to yield greater performance improvements even without increasing the response length, highlighting the potential for continuous enhancements in iterative preference optimization. The results on Llama-3-8B-Instruct and Gemma-2-9B-It demonstrate that, due to our iterative training pipeline and the AIPO training objective, we achieve state-of-the-art performance across all benchmarks compared to other iterative preference optimization methods. Additionally, length exploitation is effectively controlled.

## 5.3 COMPARISONS WITH SCALED-UP LLMS

To facilitate comparisons with the most advanced proprietary models, we scaled our base model from 13B to a maximum of 123B. As shown in Tab. 5, our 13B model, Mistral-Nemo-Instruct-2407-AIPO, achieves a 12% improvement in win rate on Arena-Hard compared to the base model, without any increase in average response length. Gemma-2-27B-It-AIPO and Llama-3-70B-Instruct-AIPO achieve improvements in win rates of 8.3% and 16.9% respectively on Arena-Hard, with about 8% and 4% of increasements in response length. When training the 123B base model, Mistral-Large-Instruct-2407, we managed to achieve an additional 12.2% improvement of win rate, despite the base model already attaining a 70.4% win rate, while the average response length is shorter than GPT-4 Omni (05/13) and GPT-4o Mini. These results highlight the effectiveness of our approach in balancing performance gains with response length in preference optimization, further demonstrating the scalability and robustness of iterative training pipeline and AIPO training objective across different model sizes.

## 6 CONCLUSION

In this study, we investigate the transition from non-iterative DPO training to iterative alignment of LLMs using synthetic data. Our research addresses a significant gap in the existing literature by providing a comprehensive comparison and analysis of iterative preference optimization training methodologies and presenting a robust framework for iterative preference optimization. Building upon this foundation, we examine the phenomenon of length exploitation in iterative training, which significantly degrades performance. We examine the characteristics of self-generated responses and introduce AIPO, an approach designed to incorporate agreement-aware adjustments in the training objective to mitigate the length issue and ensure stability in iterative preference optimization. By integrating these techniques, we achieve state-of-the-art performance on MT-Bench, AlpacaEval 2.0, and Arena-Hard without compromising on length.

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

# A APPENDIX

## A.1 IMPLEMENTATION DETAILS

**Base Model**  In our experiments, we use Mistral-7B-Instruct-v0.2 (Jiang et al., 2023a) as the base model for investigating synthetic data curation and iterative training in Sec. 3.1 and 3.2 due to limited computation resources. We then use Mistral-Nemo-Instruct-2407 (Mistral AI team, 2024b), a more advanced LLM, in Sec. 4 for developing our training objective to demonstrate the capability of our method. Note that Mistral Nemo is a 12B model, which acts as a drop-in replacement for Mistral 7B with more capable performance. To provide a detailed comparison with other iterative training methods and proprietary models, we trained on Llama 3 (Dubey et al., 2024), Gemma 2 (Team et al., 2024) and Mistral-Large-Instruct-2407 (Mistral AI team, 2024a), as shown in Tab. 4.

**Training Data**  We use UltraFeedback as the data source for all experiments. UltraFeedback is a large-scale preference dataset containing approximately 64K prompts from various sources. The responses in UltraFeedback are generated by multiple LLMs and annotated by GPT-4 based on four different aspects: instruction-following, truthfulness, honesty, and helpfulness. There is also a binarized version[2], which is created by selecting the highest score as the chosen response and one of the remaining as the rejected response. For training directly on UltraFeedback (row 1 in Tab.1), we utilize the binarized version. In contrast, for ranking responses on UltraFeedback with PairRM (row 2 in Tab.1), we employ the instructions and responses from the original UltraFeedback dataset. In the remaining experiments, we focus solely on the instructions from UltraFeedback, omitting the responses altogether.

**Hyperparameters**  For both data curation and training, we set the maximum prompt length to $512$ tokens and the maximum response length to $2048$ tokens. For data curation ablations in Sec. 3.1, we train on 60K preference pairs sourced from either UltraFeedback or synthetic data, with a batch size of $128$. For iterative training in Sec. 3.2 and 4, we set the batch size to $256$. After performing ablations for iterative training in Sec. 3.2, we chose to train on 20K preference pairs per iteration by default for the iterative trainnig in Sec. 4. For all experiments, we train for one epoch for each training stage using the AdamW optimizer with a learning rate of $5\mathrm{e}{-7}$. We apply a cosine learning rate schedule with $10\%$ warmup.

**Evaluation**  We evaluate on three benchmarks: MT-Bench Zheng et al. (2023b), AlpacaEval 2.0 Dubois et al. (2024); Li et al. (2023) and Arena-Hard Li et al. (2024). **MT-Bench** has 80 questions across 8 categories, with responses rated on a 10-point scale. We report the average scores assigned by GPT-4-Turbo. **AlpacaEval 2.0** includes 805 questions, with GPT-4-Turbo acting as both baseline and judge. It calculates win rate against baseline and includes a length-controlled win

---

[2]https://huggingface.co/datasets/HuggingFaceH4/ultrafeedback_binarized

Table 6: Effect of rule-based filtering on training with synthetic data.

| Filtering | Arena-Hard | | AlpacaEval 2.0 | | | MT-Bench |
|---|---|---|---|---|---|---|
| | WR (%) | Avg. Token | LC (%) | WR (%) | Avg. Len | GPT-4-Turbo |
| w/o Filtering | 19.0 | 679 | 25.5 | 27.9 | 2250 | 6.3 |
| w/ Filtering | **19.6** | 615 | **26.0** | **28.2** | 2130 | **6.4** |

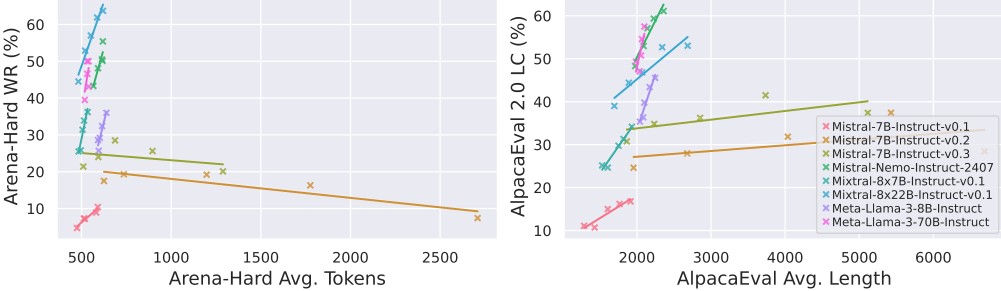

Figure 5: The ablation of base model for iterative preference optimization.

rate, which aims to mitigate the impact of length gameability of the LLM judge. We report the win rate (WR), the length-controlled win rate (LC), and the average character length (Avg. Len). **Arena-Hard** is an improved version of MT-Bench, featuring 500 high-quality prompts selected from user queries. GPT-4 (03/14) is used as the baseline model, with GPT-4-Turbo serving as the annotator. It calculates the win rate against the baseline model. We report the win rate (WR) and average token number (Avg. Token).

## A.2 FILTERING FOR SYNTHETIC PREFERENCE DATA

Our training pipeline relies heavily on synthetic data, making it sensitive to the characteristics of the base model, such as response style, diversity, and capacity. For smaller-scale models, the synthetic data often contains more biases and unexpected responses due to their limited capacity. To address this, we implement some basic rule-based filtering and data cleaning strategies during the self-instruct creation and candidate response generation stages to stabilize training and mitigate biases. This includes removing self-asked instructions, filtering out URLs from responses, and eliminating excessively long responses. Following the experimental settings in Sec. 3.1, we conduct experiments to verify the effectiveness of filtering. As shown in Tab. 6, applying filtering slightly improves overall performance. We thus apply filtering by default for all experiments in the main paper.

## A.3 MODEL ABLATION FOR ITERATIVE PREFERENCE OPTIMIZATION

To investigate how the base model affects length issue in iterative preference optimization, we perform model ablation in iterative settings, and employ vanilla DPO as the training objective in the training phase. The results are shown in Fig. 5. Our conclusion is that a stronger base model generally experiences fewer length issues. At the same time, length exploitation also appears to be influenced by training methods and training data. For example, both Mistral-7B-Instruct-v0.2 and Mistral-7B-Instruct-v0.3 exhibit severe length problems during iterative preference optimization. The pretraining and instruction tuning of LLMs is a broad topic that extends beyond the scope of this work. Additionally, many details of the training processes are often opaque. Therefore, we only highlight the potential impact of the base model based on our observations.

## A.4 STUDYING THE RELATION OF AGREEMENT-AWARENESS TO LENGTH REGULARIZATION

We perform a statistical analysis of the response lengths and log probabilities, the results are shown in Fig. 6a. The results indicate that the response length has a strong correlation with log probabilities: as the response length increases, the log probabilities tend to decrease. This is due to the inherent diversity of the model during the generation process.

(a) The distribution of response lengths and the corresponding log probabilities by the reference model.

(b) The distribution of $s_{\text{ref}}$ and the difference in length between chosen and rejected responses.

Figure 6: Statistical analysis for the training stage.

Examining the gradient weight in Eq. 9, we observe that after expressing $s_{\text{ref}}$ as the difference in log probabilities of the reference model for chosen and rejected responses, the gradient weight can be written as:

$$w_\theta = \sigma\big(\beta\big(s_{\text{ref}} - s_\theta + \alpha\big(\log \pi_{\text{ref}}(y_w \mid x) - \log \pi_{\text{ref}}(y_l \mid x)\big)\big)\big). \tag{12}$$

When combined with the relationship between log probabilities and response length, Eq. 12 appears to resemble the length regularization used in R-DPO (Park et al., 2024). The additional term $\alpha\beta \cdot s_{\text{ref}}$ gives a higher gradient weight when the chosen response is shorter than the rejected one, and a lower gradient weight when the chosen response is longer. This suggests the need to further analyze the relationship between $s_{\text{ref}}$ and the length difference $|y_w| - |y_l|$. In Fig. 6b, we plot the distribution of $s_{\text{ref}}$ and $|y_w| - |y_l|$. We can see that although there is some correlation between $s_{\text{ref}}$ and the length difference, its value is lower than that presented in Fig. 6a (from $\rho = -0.69$ to $\rho = -0.47$). This is mainly because AIPO takes into account the preferences of the reference model, which explains why alpha-DPO and AIPO perform better than R-DPO, as shown in Fig. 4.

## A.5 HYPERPARAMETER TUNING

Previous work suggests that the choice of hyperparameters is crucial for the training of non-iterative preference optimization models (Meng et al., 2024). We also observe the same phenomenon in iterative preference optimization. To ensure a fair comparison and analyze the impact of hyperparameters in $\alpha$-DPO and AIPO, we perform detailed hyperparameter tuning for the experiments presented in Sec. 4. We detailed the range of hyperparameter search for each method in Tab. 7. As depicted in Fig. 3, the response length on the AlpacaEval 2.0 benchmark continuously increases across all methods with each training iteration. This trend facilitates the examination of length exploitation in iterative preference optimization by concurrently considering the win rate on AlpacaEval 2.0 and response length. We thus compare the tend of length-controlled win rate growth with response length on AlpacaEval 2.0, as shown in Fig. 7. For each method, we select the best-performing model (positioned in the top left corner of the figure) to be included in the main paper, considering its performance on AlpacaEval 2.0 while taking into account the response length.

Fig. 7a illustrates that although a higher beta value limits the increase in length with training iterations, it also restricts performance, thereby failing to improve the model's efficiency in utilizing the response length. Similarly, augmenting DPO with the NLL term also limits both the increase in length and performance concurrently, as shown in Fig. 7b, thus proving ineffective in resolving length exploitation in iterative settings. Both adjustments appear to constrain the increase in length with respect to training iteration but leave performance unchanged for equivalent response length,

Table 7: The range of hyperparameter search for each training objective.

| Method | Objective | Hyperparameter |
|---|---|---|
| **DPO** (Rafailov et al., 2024) | $-\log \sigma\left(\beta \log \frac{\pi_\theta(y_w\mid x)}{\pi_{\text{ref}}(y_w\mid x)} - \beta \log \frac{\pi_\theta(y_l\mid x)}{\pi_{\text{ref}}(y_l\mid x)}\right)$ | $\beta \in [0.1, 0.5, 1.0]$ |
| **DPO+NLL** (Pang et al., 2024) | $-\log \sigma\left(\beta \log \frac{\pi_\theta(y_w\mid x)}{\pi_{\text{ref}}(y_w\mid x)} - \beta \log \frac{\pi_\theta(y_l\mid x)}{\pi_{\text{ref}}(y_l\mid x)}\right) - \frac{\alpha}{\|y_w\|}\log\left(\pi_\theta(y_w\mid x)\right)$ | $\beta \in [0.1]$ $\alpha \in [0.2, 0.4, 0.6]$ |
| **SimPO** (Meng et al., 2024) | $-\log \sigma\left(\frac{\beta}{\|y_w\|}\log \pi_\theta(y_w\mid x) - \frac{\beta}{\|y_l\|}\log \pi_\theta(y_l\mid x) - \gamma\right)$ | $\beta \in [2.0, 3.0]$ $\gamma \in [0.5, 1.0]$ |
| **R-DPO** (Park et al., 2024) | $-\log \sigma\left(\beta \log \frac{\pi_\theta(y_w\mid x)}{\pi_{\text{ref}}(y_w\mid x)} - \beta \log \frac{\pi_\theta(y_l\mid x)}{\pi_{\text{ref}}(y_l\mid x)} + (\alpha\|y_w\| - \alpha\|y_l\|)\right)$ | $\beta \in [0.1, 0.5]$ $\gamma \in [0.001, 0.002]$ |
| **$\alpha$-DPO** | $-\log \sigma\left(\beta \log \frac{\pi_\theta(y_w\mid x)}{\pi_\theta(y_l\mid x)} - (1+\alpha)\beta \log \frac{\pi_{\text{ref}}(y_w\mid x)}{\pi_{\text{ref}}(y_l\mid x)}\right)$ | $\beta \in [0.1]$ $\alpha \in [0.04, 0.05, 0.06, 0.07, 0.08]$ |
| **AIPO** | $-\log \sigma\left(\beta \log \frac{\pi_\theta(y_w\mid x)}{\pi_\theta(y_l\mid x)} - (1+\alpha)\beta \log \frac{\pi_{\text{ref}}(y_w\mid x)}{\pi_{\text{ref}}(y_l\mid x)}\right) - \frac{\lambda}{\|y_w\|}\log\left(\pi_\theta(y_w\mid x)\right)$ | $\beta \in [0.1]$ $\alpha \in [0.03, 0.05, 0.07, 0.09]$ $\lambda \in [0.2, 0.5]$ |

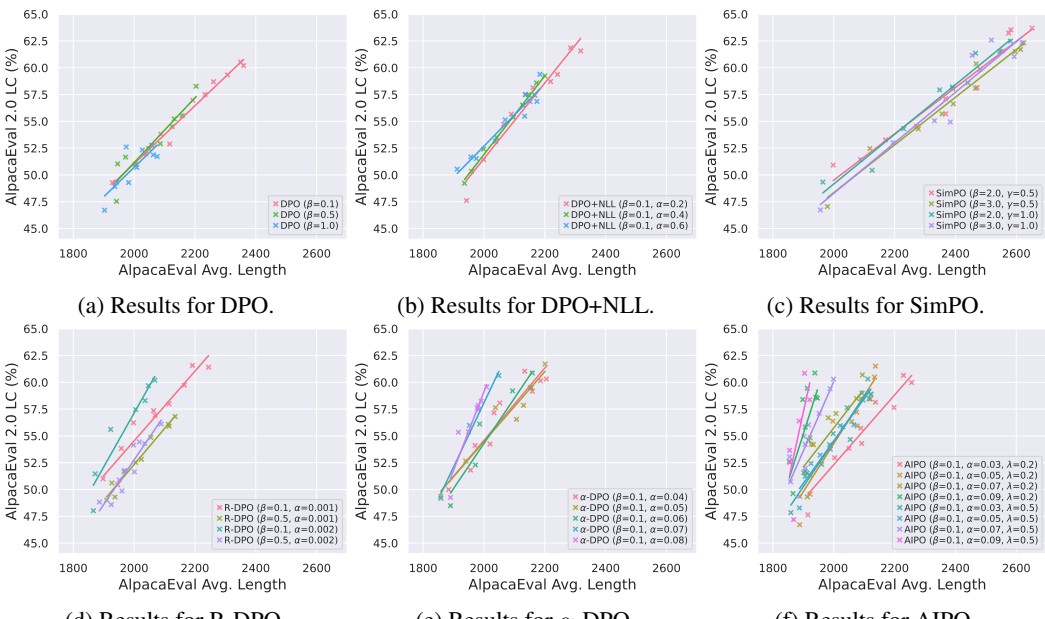

(a) Results for DPO.  (b) Results for DPO+NLL.  (c) Results for SimPO.

(d) Results for R-DPO.  (e) Results for $\alpha$-DPO.  (f) Results for AIPO.

Figure 7: The results of hyperparameter tuning for each method.

thus having no impact on mitigating the length issue. Additionally, modifying the $\beta$ and $\gamma$ values in SimPO does not affect length efficiency either. Unlike DPO, SimPO lacks a reference model, hence adjusting the $\beta$ value has no effect on the increase in response length. Conversely, R-DPO, which directly regularizes based on response length, performs more effectively in this scenario by improving performance for equivalent response lengths. Although $\alpha$-DPO does not directly target response length, it serves as a good alternative to R-DPO by further reducing response length without compromising the upper bound of performance. The results in Fig. 7e and 7f indicate that the value of $\alpha$ acts as a proxy for controlling response verbosity, where a higher $\alpha$ value leads to more concise responses, resulting in a higher win rate under equivalent lengths. However, a stability issue arises with $\alpha$-DPO due to the decrease in implicit reward value. As discussed in the main paper, we incorporate the NLL term to further enhance performance. Fig. 7f demonstrates that the addition of the NLL term significantly enhances performance at the same response length when $\alpha$ is high, while reducing length fluctuations during training.

