# OpenReview forum: "AIPO: Agreement-Aware Iterative Preference Optimization for Length Exploitation Mitigation"
_ICLR.cc/2025/Conference — ICLR 2025 Conference Withdrawn Submission_

### Official Review · Reviewer_CPqL · 2024-10-26

**Soundness:** 2
**Presentation:** 2
**Contribution:** 1
**Rating:** 3
**Confidence:** 4

**Summary:**

The research question addressed in this paper pertains to the length utilization challenge encountered in the alignment of Language Learning Models (LLMs), particularly when employing the DPO method for iterative preference optimization. The authors have recognized the gravity of this issue and have proffered an innovative solution, AIPO, which represents an intriguing research avenue worth delving into.

**Strengths:**

1.    The article conducts a thorough investigation into the length exploitation issue within the current field of LLM alignment, which is a significant and underexplored challenge with practical implications.
2.    The authors introduce AIPO, a novel training objective designed to mitigate the issue of length exploitation in iterative preference optimization, showcasing the study's originality and innovative spirit.
3.    Extensive experimentation has been conducted across datasets.

**Weaknesses:**

1.    The author posits the first contribution as a Synthetic Data Curation Pipeline for Preference Optimization, which contrasts Synthetic Data with Human-Generated data. However, it is noted that the comparison does not incorporate a blend of both types of data. There is no doubt regarding the validity of the author's experimental procedures; rather, I entertain concerns regarding the potential detrimental effects of employing exclusive Synthetic Data. A more judicious approach might be the inclusion of hybrid data. Could the author present outcomes derived from utilizing a mixture of synthetic and human-generated data?
2.    I deeply commend the author's endeavors to refine the iterative DPO approach. However, the depiction in Figure 3 astonishes me. It appears that AIPO exhibits the poorest performance, while Sim-PO achieves the most superior outcomes. Am I missing something? I would kindly request the author to clarify this matter.
3.    In Table 4, it is evident that AIPO frequently falls short of SPPO, with a staggering 10% decline in performance on platforms such as Mistral-7B-Instruct-v0.2, UltraFeedback, and PairRM's AlpacaEval 2.0. Such a precipitous drop is unacceptably high, and I would be interested in an elucidation from the authors regarding the rationale behind this substantial 9.7% disparity in favorability between AIPO and SPPO in this specific context.
4.    The methodology presented in Table 5 is inadequately expounded. For instance, the evidence is clearly manifest in Table 4 that SPPO occasionally outperforms AIPO. Yet, Table 5 omits any discussion of this matter. Might it be the case that the 123B Mistral-Large-Instruct-2407 model demonstrates superior performance with SPPO over AIPO?
5.    The authors offer a meager synthesis of preference alignment methodologies, with the present article notably lacking a comprehensive summary and description of pertinent prior work. This deficiency hinders the understanding of the broader preference alignment field by other researchers. I suggest the author supplement this with a more extensive synthesis in the form of a survey and consider the citation of following literatures:

[1]	Wang Z, Bi B, Pentyala S K, et al. A Comprehensive Survey of LLM Alignment Techniques: RLHF, RLAIF, PPO, DPO and More[J]. arXiv preprint arXiv:2407.16216, 2024.

[2]	Shen T, Jin R, Huang Y, et al. Large language model alignment: A survey[J]. arXiv preprint arXiv:2309.15025, 2023.

[3]	Azar M G, Guo Z D, Piot B, et al. A general theoretical paradigm to understand learning from human preferences[C]//International Conference on Artificial Intelligence and Statistics. PMLR, 2024: 4447-4455.

[4]	Wang C, Jiang Y, Yang C, et al. Beyond reverse kl: Generalizing direct preference optimization with diverse divergence constraints[J]. arXiv preprint arXiv:2309.16240, 2023

[5]	Sun H, Zheng Y, Zhao Y, et al. Generalizing Offline Alignment Theoretical Paradigm with Diverse Divergence Constraints[C]//ICML 2024 Workshop on Models of Human Feedback for AI Alignment. 2024.

[6]	Chen H, Zhao H, Lam H, et al. Mallows-DPO: Fine-Tune Your LLM with Preference Dispersions[J]. arXiv preprint arXiv:2405.14953, 2024.

[7]	Rafailov R, Hejna J, Park R, et al. From $ r $ to $ Q^* $: Your Language Model is Secretly a Q-Function[J]. arXiv preprint arXiv:2404.12358, 2024.

Generally speaking, the paper's proposed methodology is innovative, presenting an intriguing perspective; however, I find the efficacy of this method profoundly perplexing. Thus, the authors should elucidate the issues surrounding the 'weakness' aspect. Should the authors address my concerns, I would be inclined to enhance my rating.

**Questions:**

See 'Weaknesses' Part.

---

### Official Review · Reviewer_cYa5 · 2024-11-03

**Soundness:** 2
**Presentation:** 2
**Contribution:** 2
**Rating:** 3
**Confidence:** 4

**Summary:**

This paper introduces Agreement-Aware Iterative Preference Optimization (AIPO), a novel training objective designed to address the issue of length exploitation in Direct Preference Optimization (DPO) for Large Language Models (LLMs). By rethinking the standard DPO loss, AIPO incorporates agreement-aware adjustments to reduce length bias in iterative preference optimization tasks, where synthetic data is used iteratively to align models with user preferences. The approach is evaluated on MT-Bench, AlpacaEval 2.0, and Arena-Hard, where AIPO demonstrates significant performance gains while effectively controlling response length

**Strengths:**

1. Addressing Length Exploitation: The paper tackles a relevant issue in DPO, where longer responses are often favored in preference optimization, leading to inefficiencies. AIPO’s approach to mitigating this bias is well-motivated and practical, making it useful for applications where response length is critical.

2. Iterative Training with Synthetic Data: The use of synthetic data for iterative training in AIPO showcases a scalable alternative to human-labeled data. The approach is thorough, covering instruction generation, candidate response ranking, and preference optimization, which provides a complete pipeline for training without additional human intervention.

3. Good Results: AIPO achieves notable improvements on benchmarks like AlpacaEval 2.0 and Arena-Hard, which validates its effectiveness over conventional DPO and other preference optimization methods.

**Weaknesses:**

1. AIPO introduces three additional hyperparameters that must be manually adjusted to achieve optimal performance, significantly increasing the complexity and time cost of experiments. This greatly limits the potential application of this method for large language models in real-world settings.

2. The technical contributions are quite limited. Similar ideas of introducing a margin and adding SFT regularization have already been proposed by many previous works [1, 2, 3] and [4, 5].

3. The experimental evaluation is weak. The authors only conduct experiments on instruction-following benchmarks, which are quite sensitive to sampling and decoding hyperparameters. The authors should evaluate models trained with different methods on various tasks listed on the Hugging Face Open Leaderboard and datasets with safety and honesty metrics, such as HH-RLHF.

[1] SimPO: Simple Preference Optimization with a Reference-Free Reward. NeurIPS 2024

[2] Disentangling Length from Quality in Direct Preference Optimization. ACL Findings 2024

[3] Direct Preference Optimization with an Offset. ACL Findings 2024

[4] Contrastive Preference Optimization: Pushing the Boundaries of LLM Performance in Machine Translation. ICML 2024

[5] Iterative Reasoning Preference Optimization. NeurIPS 2024

**Questions:**

Please see above

---

### Official Review · Reviewer_F7hd · 2024-11-04

**Soundness:** 2
**Presentation:** 1
**Contribution:** 2
**Rating:** 5
**Confidence:** 3

**Summary:**

This paper takes a deeper look at the problem of length exploitation when training LLMs with DPO, showing that iterative self-training with synthetic data leads to better results, but with longer self-generated responses than seen in the human generated reference distribution. The authors then propose a modification to the DPO loss aimed at improving iterative preference optimization. The authors validate that approach AIPO leads to improved performance on MT-Bench, AlpacaEval, and Arena-Hard with shorter self-generated responses.

**Strengths:**

I think this paper explores an interesting topic. It is definitely important to understand the learning properties of DPO with self-generated responses. Moreover, a strength of this paper is the empirical results. Sometimes they are not super convincing, but the paper considers a a solid set of benchmarks and compares to some logical baselines for the most part.

**Weaknesses:**

The writing quality is a significant weakness of this paper. There are numerous grammatical errors throughout. Moreover, the authors do a very poor job of discussing their contributions. For example, in the introduction three contributions are listed, but the novelty of each of these contributions is entirely unclear based on the description.  The term "length exploitation" is used throughout but is never explained formally, which is odd because it appears to be the central motivation of this paper.  Honestly, even in the empirical results discussing it, I do not find the differences in length to be that egregiously large, so I didn't even really come away with thinking that it is the central problem these approaches need to tackle -- it seems more like a potentially interesting byproduct.  Also, the motivation for the particular approach was very handwavy with no obvious connection to the concept of length exploitation. Really it seems like the approaches considered to address length exploitation do so indirectly, which means that length exploitation is not really the central problem. If this is the case, then the entire framing of the paper is written poorly because you are really asserting with AIPO that the problem is something else and lack length exploitation is a potentially nice side effect.

While I think there very well may be some merit to the approach proposed by the authors, I just feel that there are too many modifications needed to make the paper clear for the ICLR audience for me to support acceptance.

**Questions:**

Q1: I found the phrasing of your first contribution in the introduction related to "Data" very confusing. Which aspects of this contribution are novel and how does this relate to findings in previous papers?

Q2: With respect to the second contribution related to "Finding" could you please highlight what aspects of this finding are novel and which already exist in the literature?

Q3: With respect to the third contribution related to "AIPO" could you explain which aspects of AIPO are novel i.e. in comparison to approaches that already exist like R-DPO and SimPO. Especially looking at Table 7, I am aware of the literal differences, but am trying to get the authors to really frame their contribution i.e. we are the first to do XXX so that the relative novelty is clear and readers can differentiate between conceptual contributions and specific implementation choices.

Q4: Could you provide a formal description of the "length exploitation" problem?

Q5: Looking at the description of the SimPO loss in the appendix, it seems like some key differences are that a reference model and policy ratios are not used. However, the explanation of how this model performs in the experiments section largely focuses on the length normalization aspect. Do you have a reason to think this length normalization part specifically does not work?

Q6: The authors write "We hypothesize this makes the DPO training objective susceptible to self-generated preference pairs, consequently degrading the model’s learning and resulting in responses that are lengthy and less informative." Where do the lengthy and less informative assertions come from here? Aren't there other possible results of degraded learning?

---

### Note · Authors · 2024-11-28

I have read and agree with the venue's withdrawal policy on behalf of myself and my co-authors.